# Bimaxillary Distalization with Skeletal Anchorage for Management of Severe Skeletal Class III Open Bite Malocclusion

**DOI:** 10.3390/children9111666

**Published:** 2022-10-31

**Authors:** Abdulrahman Alshehri

**Affiliations:** Division of Orthodontics, Department of Preventive Dental Sciences, Faculty of Dentistry, Jazan University, Jazan 45142, Saudi Arabia; aalshehri@jazanu.edu.sa

**Keywords:** Angle’s class III malocclusion, distalization, miniplates, open-bite, orthognathic surgery, skeletal anchorage

## Abstract

Adult patients with severe vertical growth pattern, skeletal class III malocclusion and open bite anteriorly are difficult orthodontic cases to treat. An orthognathic surgical treatment plan may benefit adult patients with such types of complex malocclusions, however in certain cases, the patient’s medical history may contraindicate the surgical treatment plan. A male patient aged 17 years presented with a prognathic mandible, Angle’s class III malocclusion on a class III skeletal base with proclination in upper incisors, retroclination of lower incisors, and reduced facial convexity. Patient gave history of asthma and complex cardiac diseases including arrhythmia, irregular heartbeat, and pacemaker. This case report describes a non-surgical approach of distalization of mandibular and maxillary arches performed with the help of miniplates to achieve an improvement in the facial balance without surgery.

## 1. Introduction

Class III malocclusions, when associated with hyperdivergent growth pattern and anterior open bite, are the most difficult orthodontic cases to treat in the late adolescent and adult patients [1,2]. Adult patients with significant skeletal Class III discrepancy tend to have mandibular asymmetry and reduced condylar translational movement [3]. Maxillary advancement and/or mandibular setback surgery may be required to correct the anteroposterior skeletal relation and improve the mandibular movement and the mandibular deviation during mouth opening [4]. However, a non-surgical alternative is required when orthognathic surgery is contraindicated due to patient’s medical conditions, or if the patient refuses orthognathic surgery due to the potential risks and complications involved with anaesthesia and the surgical procedure. 

The treatment alternative for management of such malocclusion is orthodontic camouflage for patients not wanting to undergo orthognathic surgery. Orthodontic camouflage can be undertaken with or without extractions depending on how severe the skeletal discrepancy along with facial profile [5]. Several extraction schemes have been used for orthodontic camouflage including extraction of four premolars, two mandibular premolars or mandibular incisor [6]. Another way of orthodontic camouflage approach is to distalize the lower molars with a non-extraction treatment plan) [7]. 

Distalization of both the upper and lower molars is a challenging tooth movement, especially in adult patients. Certain appliances, such as the head-gear appliance anchored to the mandibular teeth, lip bumper, and franzulum-appliance, have been used traditionally for distalization of mandibular molars. However, there are certain limitations for such appliances as they require a patient’s compliance and may lead to proclination of anterior teeth. In addition, a poor control of the force vector with such appliances can lead to extrusion forces on the lower posterior teeth and increase the vertical dimension with downward and backward rotation of mandible, which can be detrimental in cases with open-bite in the anterior region. 

The force vectors can be controlled in a more efficient manner for effective distalization of mandibular arch, as compared to the conventional appliances, after the temporary anchorage devices (TADs) were introduced [8]. The placement of inter-radicular miniscrews have been described for the distalization of the mandibular arch [8]. However, the failure rate was high for interradicular mini-implants, especially in the mandible, the risk of damaging the roots, and necessity of relocation of TADs to allow for additional distalization; these are the main problems associated after using interradicular mini-implants [9]. Other sites such as retromolar mini-implants and buccal-shelf mini-implants can also help in distalization of the mandibular teeth. However, the high failure rate of such mini-implants is a concern that prevents many clinicians from adopting this approach [7].

On the other hand, Titanium miniplates, defined as skeletal anchorage systems (SAS), have shown very high success rates [8]. The miniplates do not interfere with the distalization tooth movement as they are placed outside dentition; and thus, do not need to be relocated during the treatment. SAS provides efficient biomechanics and enables intrusion and en masse distalization of the mandibular and maxillary dentitions in adult patients [8]. However, the reports of using such an assembly in cases with skeletal class III open bite in the anterior region and medical conditions are scarce. 

This study presents an orthodontic camouflage treatment option for an adult male patient who had a Class III skeletal pattern, anterior open bite, and maxillary anterior crowding. Four miniplates were placed in the maxilla and mandible for distalizing the upper and lower dentition after removal of all the third molars to relieve the maxillary crowding, establishing Angle’s class I malocclusion and obtaining ideal overbite and overjet.

## 2. Diagnosis and Etiology

The patient gave a history of asthma and complex cardiac diseases including arrhythmia, irregular heartbeat, and pacemaker on medical examination. Familial history showed the patient’s father to have similar class III malocclusion. There were no signs and symptoms of dysfunctioning temporo-mandibular joint at the initial examination.

The extraoral examination showed a concave profile, protruding lower lip, average nasolabial angle and increased lower facial height. On the intraoral examination, Angle’s Class III molar relationship on the right side and a super Class I molar on the left side and Class III canine relationships bilaterally were seen. Anterior open bite of 3.5 mm with maxillary anteriors showing mild crowding of 3.5 mm was seen. Mandibular midline was deviated 2 mm to the left side (Figure 1).

All the permanent teeth were present in the panoramic radiograph along with asymmetric chin and multiple restorations. No other significant pathology was noted on the panoramic radiograph. On lateral cephalometric analysis (Table 1) skeletal Class III jaw relationship with increased mandibular plane angle was seen, and there was increase in gonial angle and lower facial height. In addition, the patient had proclined maxillary incisors, retroclined mandibular incisors and acute interincisal angle. The cervical vertebral maturation (CVM) stage showed the presence of 5–10% growth remaining. The systematic reviews conducted by Szemraj A et al. and Ferrillos M et al. showed that skeletal maturation indication by CVM Baccetti and Hassel and Farman is more reliable than the Handwrist method in growing children [10,11].

## 3. The Treatment Objectives

The treatment objectives were to distalize the mandibular teeth dentition to achieve normal overbite and overjet and improve the lower lip profile, to distalize the upper dentition, relieving the anterior crowing, correcting the maxillary and mandibular midlines, to achieve Class I molar and canine relationship improving the maxillary incisal display providing the patient with a good occlusal relationship, a pleasant smile and facial appearance.

## 4. Treatment Alternatives

The treatment alternative consisted of a combination of orthodontic treatment and orthognathic surgery with a Lefort I osteotomy and setback of mandible to correct the skeletal discrepancy and achieve maximum correction of the dental and facial esthetics. However, the surgeon decided that the patient was not a good candidate for an aggressive surgical invasion due to the potential complications associated with his cardiac diseases. Thus, because of the patient’s medical history, the surgical option was rejected. 

Extraction of four premolars as camouflage treatment was the second alternative plan. This option would enable the correction of occlusal relationships, improve overbite and overjet, correct the maxillary crowding, and the facial profile. However, the patient did not agreeing with the extraction of premolars.

The last alternative was to distalize the maxillary and mandibular arches using skeletal anchorage, which would help in achieving the treatment objectives. The non-extraction treatment approach of maxillary and mandibular arch distalization with miniplates was selected by the patient.

## 5. Treatment Progress

The treatment plan was well explained to the patient and consent was obtained. Before orthodontic treatment, the mandibular and maxillary third molars were extracted by the oral surgeon and four skeletal anchorage plates were placed. The miniplates were placed behind the mandibular and maxillary second molars on all sides and secured in place by two monocortical miniscrews (5 mm in length and 2 mm in diameter). The heads of the miniplates were positioned horizontally and laterally to that of the buccal surface of the second molar. MBT 0.022′ slot brackets were bonded on maxillary and mandibular arches. Segmental 0.016-in nickel–titanium archwires were used for alignment and leveling of the maxillary and mandibular dentition and continued up to 0.019 × 0.025-in stainless steel in 4 months. From the upper and lower miniplates to the first premolars using elastomeric chains on 0.019 × 0.025-in stainless steel sectional wires, force of approx. 200 g was applied (Figure 2A). 

The distalization in both the upper and lower arches was discontinued after four months when the space of about 1.5 mm distal to maxillary canine was achieved in order to correct the crowding, along with achieving a Class I molar relationship bilaterally. Subsequently, the bonding of the brackets was done on the remaining teeth in both the arches and sequential wire progression was performed (Figure 2B). The spaces between canine and first premolar were closed in 0.017 × 0.025-in stainless steel wire with the help of elastomeric chains from the miniplates (Figure 2C). After correcting the anterior open bite and achieving Class I molar and canine relationships, finishing and detailing was done with finishing bends in 017 × 025-in Connecticut new archwire (CNA).

The total orthodontic treatment time was 25 months. The miniplates were stable throughout the period of orthodontic treatment and were removed after debonding the orthodontic appliances. Maxillary Hawley retainer and mandibular lingual fixed retainer were used for retention. 

## 6. Treatment Results

The patient was quite satisfied with the orthodontic outcome and facial profile. The posttreatment facial photographs showed an improvement in the facial balance due to retraction of the lower lip (Figure 3). Class I canine and molar relationships on both sides, normal overbite and overjet, and proper alignment were achieved, as seen in Figure 3. The dental midline was improved, and the unilateral posterior crossbite was corrected. In terms of the transverse dimension, the maxillary intercanine width was increased from 31.6 mm to 33.4, and the maxillary intermolar width was increased by 1.3 mm. In the mandible, no change was observed in the intercanine width, whereas the intermolar width was increased by 2.6 mm. 

Acceptable root parallelism, with no significant signs of resorption of root, was shown in posttreatment panoramic radiograph, except for upper left maxillary central incisor. There was no change in the ANB angle, and an increase in the interincisal angle was seen (Table 1). According to the superimposition, maxillary incisors showed slight extrusion, and mandibular incisors showed relative extrusion and retraction of about 3 mm with controlled tipping (Figure 4). The position of the lower lip was improved, and no other remarkable change was observed in the facial height and soft tissue profile. The movement of the mandibular and maxillary first molars could be considered almost bodily translation, because the crown of the mandibular first molar moved 3 mm distally and the roots moved 2.5 mm distally, and both the crown and roots of the maxillary first molar moved about 1 mm distally (Figure 4).

## 7. Discussion

It is necessary to realize that non-surgical alternatives are required for successful management of the severe skeletal class III patients when medical conditions contraindicate orthognathic surgery. In this patient, esthetically pleasant results were achieved by whole-arch distalization of maxillary and mandibular dentition with skeletal anchorage. The facial and occlusal changes produced as a result of treatment were directly related to the dentoalveolar compensatory changes associated with en masse distal movement of the mandibular dentition, and the occlusal plane rotating counterclockwise that was planned in the treatment goals for the patient.

The applications of mini-implants have been expanded to involve more complex tooth movements such as maxillary expansion, extrusion, intrusion of posterior segments, and distalization [8,12]. Previous studies have shown that the amount of maxillary and mandibular dentitions distalization ranged from 3 to 4 mm using different types of skeletal anchorage [8,13]. The lingual cortex of the mandibular body has been reported to be the limiting factor for mandibular molar distalization, whereas maxillary tuberosity is considered to be the posterior limit for maxillary distalization [13]. In this patient, maxillary and mandibular molars were distalized by 1 mm and 3 mm, respectively, with almost bodily translation. A possible explanation for this type of tooth movement could be related to the use of stiff 0.019 × 0.025-in stainless steel archwire in a slot size of 0.022-in during molar distalization, and the force applied near the center of resistance of the posterior dentition.

Extrusion of posterior teeth and an increase in the mandibular plane angle due to the distalization of molars would be undesirable side effects for the current patient because it could have increased the anterior open bite, thus worsening his facial profile. Therefore, to avoid extrusion of posterior teeth, the biomechanics of the distalization forces were calibrated in such a way that the direction of forces in the upper arch was upward and backward, while in the lower arch it was downward and backward. This intrusive component of the applied forces allowed us to maintain the position of mandibular molars while distalizing and prevent the opening of the mandibular plane (Figure 5A).

Successful treatment of the anterior open bite in adult patients is a challenging task that requires appropriate application of orthodontic biomechanics. In this patient, once the mandibular posterior segment was distalized sufficiently, the mandibular archwire was changed to 0.017 × 0.025-in stainless steel for the closure of the spaces distal to mandibular canines during retraction of the mandibular incisors, achieving greater lingual tipping of incisors. Moreover, distalization forces led to a rotation of the mandibular arch in the counterclockwise direction, thereby simultaneously decreasing the vertical dimension of the occlusion and the open bite (Figure 5B) [14]. In the maxilla, the reduced incisal display on smiling and at rest allowed for the extrusion of maxillary incisors. To avoid proclination of maxillary incisors while relieving the maxillary crowding, spaces were created distal to upper canines by en masse distal movement of upper posterior teeth.

In this patient, miniscrews could have been placed in the buccal shelf area of mandible and the infrazygomatic crest to achieve the treatment objectives. Nevertheless, the high failure rate, soft tissue irritation and the difficulty of controlling the point of action of the force prevented their use for anchorage [13,15]. Thus, miniplates were chosen as the method of skeletal anchorage for this patient because they offer the advantages of being stable under application of heavy forces used for distalization, and allow for 3-dimensional control of the tooth movement [8]. Thus, this case report shares important insights for other clinicians on how the treatment plan can be modified for a patient, considering their specific needs and conditions and the utility of contemporary orthodontic biomechanics in achieving esthetic result.

## 8. Conclusions

Distalization with miniplates can be utilized as a successful non-surgical alternative in cases having skeletal class III malocclusion and anterior open bite, where the medical conditions such as cardiac diseases preclude orthognathic surgery.

## Figures and Tables

**Figure 1 children-09-01666-f001:**
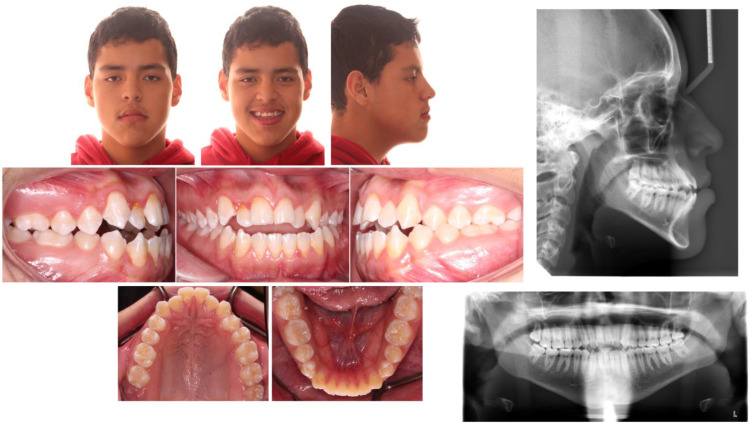
Pretreatment facial and intraoral photographs.

**Figure 2 children-09-01666-f002:**
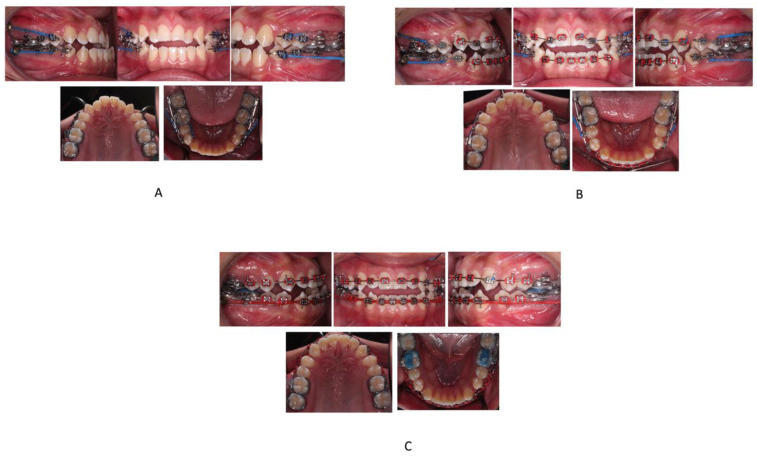
(**A**–**C**)—Progress intraoral photographs.

**Figure 3 children-09-01666-f003:**
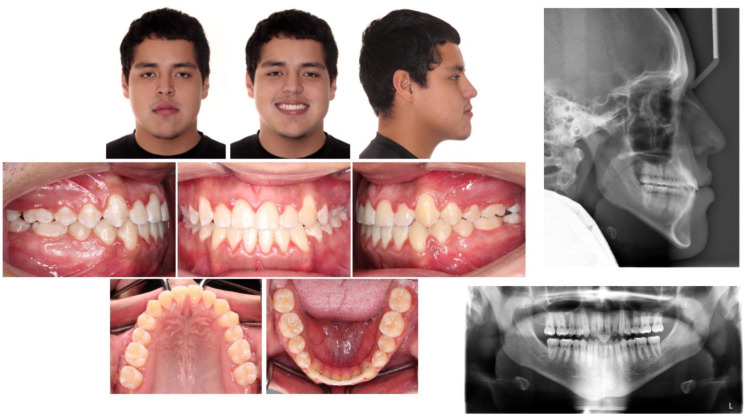
Posttreatment facial and intraoral photographs.

**Figure 4 children-09-01666-f004:**
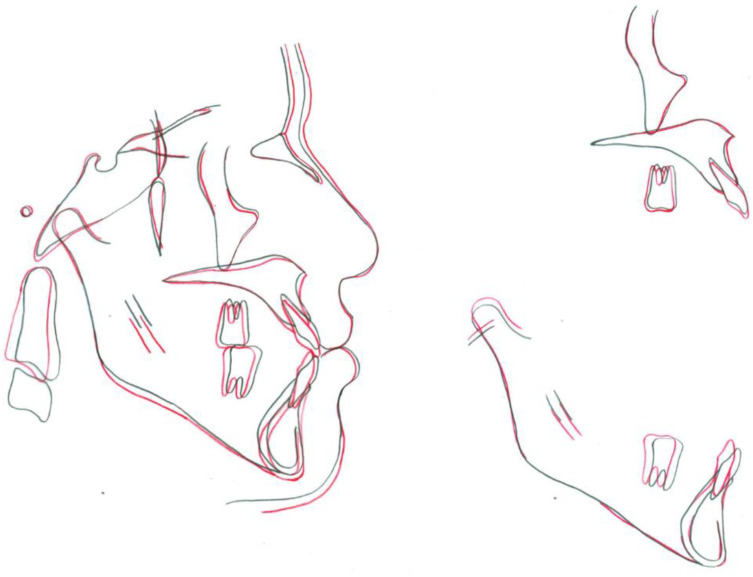
Superimpositions of pretreatment (black line) and posttreatment (red line) cephalometric tracings.

**Figure 5 children-09-01666-f005:**
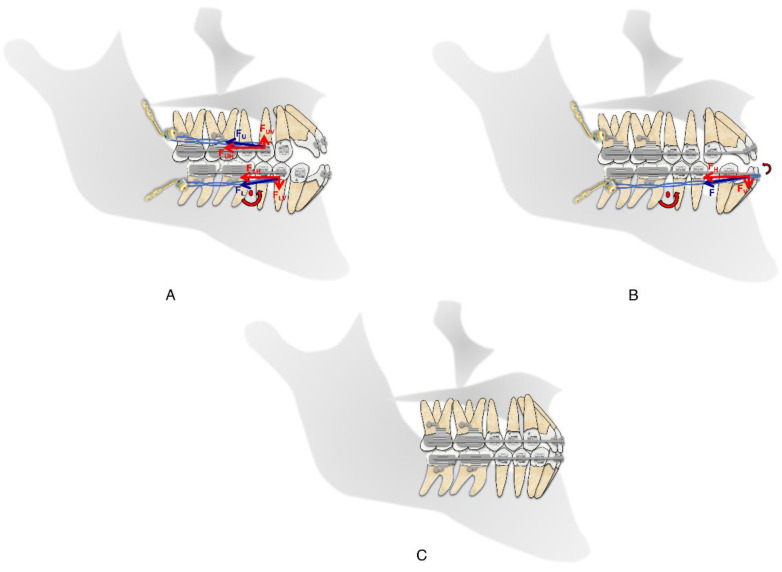
(**A**)—The directions of the applied forces on this patient were backward and upward in the maxillary arch, and backward and downward in the mandibular arch. Thus, the intrusive component of applied forces might prevent the opening of the mandibular plane. (**B**)—As the force vector is above the center of resistance of the mandibular dentition, the distalization forces lead to a counterclockwise rotation of the mandibular arch, thereby simultaneously decreasing the vertical dimension of the occlusion and the open bite. (**C**)—Final treatment result.

**Table 1 children-09-01666-t001:** Cephalometric analysis.

Measurement	Norm	Pretreatment	Posttreatment
SNA (°)	82.0 ± 2.0	86.1	86.5
SNB (°)	80.0 ± 2.0	85.6	85.9
ANB (°)	2.0 ± 2.0	0.5	0.6
Wits appraisal	−1.0 ± 1.0	−4.6	−5
FMA (°)	24.0 ± 4.5	31.6	32.1
MP-SN (°)	32.0 ± 5.0	40.7	41.3
Gonial angle (Ar-Go-Me)	122.1 ± 5.29	144	142.4
L1-MP (°)	95.0 ± 7	87.4	80.7
U1-SN (°)	102.0 ± 5.5	115.4	114.3
U1-NA (mm)	4.3 ± 2.7	6.3	6
L1-NB (mm)	4.0 ± 1.8	8.6	4.3
Interincisal angle (°)	130.0 ± 6.0	116.5	129.3
Upper lip to E-line (mm)	−4.0 ± 2.0	−4.4	−4.4
Lower lip to E-line (mm)	−2.0 ± 2.0	2.4	−0.5

## Data Availability

Not applicable.

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
