# Peer review of "Bimaxillary Distalization with Skeletal Anchorage for Management of Severe Skeletal Class III Open Bite Malocclusion"

_children, 2022, doi:10.3390/children9111666_

Round 1
Reviewer 1 Report
Dear Author,
Aim of this study was to present a camouflage treatment option for Class III adult patient having open bite in the anteriors, and upper anterior crowding. More in detail, four mini-plates were placed in the maxilla and mandible for distalizing the upper and lower dentition after removal of all the third molars to relieve the maxillary crowding, establishing Angle’s class I malocclusion and obtaining ideal overbite and overjet.
I would like to congratulate to the author and to thank him for this valid article. I am really impressed with its values, preparation etc.
The study is of scientific interest and in line with the aims of the journal.
However, there are some issues that should be addressed.
Abstract
- Line 12: Please, do not start the sentence with number.
Introduction
- Line 24: “Amongt “ Please correct.
- Line 25: “(Staudt et al., 2009; Hamamci et al., 2006)”, please remove in all the text. References should be reported as number in square brackets.
Diagnosis and Etiology
In my opinion it was very important to report that the most widely used methods to assess skeletal maturation, as the hand and wrist rx as gold standard, and the CVM by Baccetti. Two systematic reviews were conducted on this topic in the last years, and authors concluded that the CVM method shows a high level of correlation with the HWM method. Please refer to and cite: Szemraj et al. Is the cervical vertebral maturation (CVM) method effective enough to replace the hand-wrist maturation (HWM) method in determining skeletal maturation?-A systematic review. Eur J Radiol. 2018 May;102:125-128. doi: 10.1016/j.ejrad.2018.03.012. and Ferrillo et al. Reliability of cervical vertebral maturation compared to hand-wrist for skeletal maturation assessment in growing subjects: A systematic review. J Back Musculoskelet Rehabil. 2021;34(6):925-936. doi: 10.3233/BMR-210003.
References were well written.
Author Response
Please see the attachment for the corrected manuscript

Reviewer 2 Report
Introduction
Analyze these two important references about
the characteristics and changes in mandibular condylar motion in patients with skeletal Class III malocclusion.
Ugolini A, Mapelli A, Segù M, Zago M, Codari M, Sforza C. Three-dimensional mandibular motion in skeletal Class III patients. Cranio. 2018 Mar;36(2):113-120.
Ugolini A, Mapelli A, Segù M, Galante D, Sidequersky FV, Sforza C. Kinematic analysis of mandibular motion before and after orthognathic surgery for skeletal Class III malocclusion: A pilot study. Cranio. 2017 Mar;35(2):94-100.
Line 37, 38 this sentence is not clear
Line 68 We? I?
Did you obtain by your patient to publish his pictures, his face and medical history?
Author Response

(The authors gave the same response as above.)

Round 2
Reviewer 1 Report
Authors modified the text according to the suggestions.
I found this work impactful and fit well with in the scope of this journal.
In my opinion, it is suitable for publication.